# Seating Behaviour of Students before and after the COVID-19 Pandemic: Findings from Occupancy Monitoring with PIR Sensors at the UCL Bartlett Library

**DOI:** 10.3390/ijerph192013255

**Published:** 2022-10-14

**Authors:** Gizem Izmir Tunahan, Hector Altamirano

**Affiliations:** Institute for Environmental Design and Engineering, University College London, London WC1H 0NN, UK

**Keywords:** COVID-19, seat preference, library, students, seating behaviour, occupancy sensors

## Abstract

Since the first case of COVID-19 was confirmed in China, social and physical distancing has been promoted worldwide as an effective community mitigation strategy. However, our understanding remains limited regarding how students would resume their activities and use of libraries when the restrictions to manage the spread of coronavirus were lifted. Understanding students’ seating behaviour in libraries is required to guarantee that the libraries meet the needs and preferences of students and promote students’ health and well-being and satisfaction with the library. This paper aims to explore the changes in the use of study spaces before and after the pandemic. Occupancy data from the UCL Bartlett Library collected at 10-min intervals from motion sensors located underneath desks was used to assess the occupancy within the library and which was then compared to characteristics of the space. This study revealed that the COVID-19 pandemic significantly impacted students’ use of the library as well as how much time they spent there. While seats with a good combination of daylight, outdoor view and privacy were in most demand before the pandemic, distance from other students seems to be the priority after the pandemic. Students’ seating preferences appear to be also influenced by the position of desk dividers after COVID-19. Future research should focus on assessing and proposing new seating arrangements and developing strategies to promote students’ satisfaction with libraries in response to changes in students’ seating behaviours.

## 1. Introduction

Since the first confirmed case of COVID-19 was reported in March 2020, physical distancing has been promoted as one of the most critical mitigation strategies for preventing the spread of COVID-19. Cities have been placed under full or partial COVID-19 lockdowns and people have been restricted to staying in their homes to stop or slow down the spread of this disease. In addition to the “stay at home” calls by official institutions, everyone was in many countries requested to wear masks in indoor and outdoor areas, especially in crowded or poorly ventilated locations, unless exempt or have a reasonable excuse for not wearing a mask such as being younger than 2 years old, having trouble breathing or unable to place or remove the mask [1], and maintain at least 1 m of social distance [2]. Studies on COVID-19 have demonstrated that this disease has profoundly changed individuals’ lifestyles; posing a challenge to interpersonal and community interactions, creating an environment of fear, anxiety and stress among societies [3], and decreasing physical health with the reduced levels of physical activity [4]. Consequently, it caused a remarkable change in individuals’ habits and lifestyle-related behaviours and choices. For instance, a study conducted in India to assess the lifestyle-related behaviour of people following COVID-19 showed that 75% of participants reported an increase in sitting and screen time, while 20% reported an increase in intake of unhealthy foods [5]. Similarly, a cross-sectional survey conducted in India to assess changes in lifestyle-related behaviour after the pandemic found that individuals’ weight gain increased as a result of changing eating behaviour, decreasing physical activity and increase in screen and sitting time [6].

In the built environment, the required physical distancing recommended during the pandemic raised concerns regarding how this disease will affect occupants’ perception and use of space and how the design of public spaces will need to be adapted [7]. Although several researchers proposed engineering solutions to improve air quality through natural or mechanical ventilation, others proposed the redesign of layouts as a simple and inexpensive way to guarantee minimum distance and maximum occupant capacity [8].

Within this context, most academic institutions have strived to ensure the continuity of the education process and developed COVID-19 safety guidelines that include seating distributions for their indoor spaces using specific restrictions and distance between desks [8]. As an essential part of an academic institution, libraries play a significant role in students’ learning process by providing an environment that enhances their learning experience and contributes to their academic and intellectual development [9,10]. Seating that meets the needs and preferences of students can promote a longer stay in the libraries and keep students motivated, which in turn influences their emotions and learning abilities [11].

Individuals’ relationship with the space, or the way of using the indoor environment, changed during the pandemic [7] and the use of libraries is also predicted to change. Hence, a better understanding of these spaces is required to guarantee [12] the students’ health and well-being and satisfaction with the library. The indoor environment can influence almost every aspect of our lives, from sleep and energy levels to mood and how productive we are. For instance, poor indoor quality has been linked to cognitive impairment, stress, depression, and even suicidal tendency, whereas exposure to proper lighting can help improve mental conditions, reduce stress and anxiety, and improve mood and productivity [13]. Creating environments that meet students’ needs and expectations is key for their mental health and well-being, especially after a period like a pandemic, which in the UK, has caused an increase in cases of students’ depression from 30 to 44% and anxiety from 22 to 27% [14], while worldwide an increase of 25% in the prevalence of anxiety and depression [15]. 

However, it is unclear how students have resumed their activities, and use of libraries after the restrictions imposed to prevent the spread of the coronavirus were lifted. This paper aims to examine the effect of the pandemic on students’ library usage using occupancy monitoring data and investigate how parameters that seemed to be key when selecting a desk became less important. Hence, the following research question has been investigated: How has the use of the library changed after COVID-19, and how has this been reflected in students’ behaviour? 

This study has helped to understand the effect of COVID-19 on students’ seating behaviours and provided insights for managers and daily operators of university buildings on the factors to be considered to guarantee the mental health and productivity of students but also help to ensure libraries are used more efficiently and, ultimately, reduce unnecessary energy consumption. This research paper briefly describes the literature on seating selection, presents the field site and how occupancy monitoring data was analysed, and finally, depicts the findings of the study; changes in the number of students using the library and the spaces most in demand before and after COVID-19.

## 2. Literature Review

The expectation of occupants and their behaviour in the built environment could vary depending on the building type, design features, climatic conditions, type of activity [16], and people’s personalities [17]. Understanding occupants’ behaviour and their interactions with the indoor environment could provide insights into how to improve occupants’ satisfaction [18] and the energy efficiency of a building [19,20]. For instance, understanding the reasons behind selecting a particular seat in an environment could help develop strategies to improve occupants’ satisfaction and maximise the benefit of an environment such as a library that has an essential role in enhancing students’ cognitive abilities and achievements.

The seat selection process results from the individuals’ prior experiences in a space or a deliberate choice among alternatives while entering the space [21], regardless of whether deciding consciously or unconsciously [22]. Seating preferences differ depending on whether a person is familiar or unfamiliar with the physical layout of a space [23]. The human response to the physical environment is strongly associated with prior experiences [24]. For example, library users could repeatedly choose the same seat depending on prior experiences, whereas first comers must rely on external sources such as existing lighting conditions, noise levels, etc. The availability of seats at a particular time could also influence seat selection; individuals arriving earlier at the library have more chances to select a seat than those arriving later. Individual differences, namely arousal, motivation, and expectation, also matter in human behaviour [24], influencing the decision-making process. All these factors could make a difference in the individuals’ seat preferences.

Linking the seating behaviour of individuals with a particular stimulus in the physical environment is difficult because individuals are exposed to multiple sources of information during the seat selection process. The behavioural response to a physical stimulus in an environment is not directly associated with its magnitude, but with the interaction of people and their surroundings [24]. The factors influencing seating behaviour in the learning environment have been defined as ambient temperature, type of furniture, proximity to other occupants [25], quietness, outdoor view, privacy, social interactions such as close to friends, entrance or circulation [26], daylight [27,28], students’ degree of territoriality and seat arrangements [29]. When choosing a space, individuals value specific factors more than equally value each factor [23]. Therefore, it is impossible to associate students’ seating behaviour with only one environmental factor. However, some factors are more dominant in the decision process of students [30]. Understanding the interaction between the physical environment and the seating behaviour of students is essential for designing a functional and comfortable learning environment.

COVID-19 is expected to have a long-term impact on people’s perceptions and behaviours in various aspects. For instance, data collected through interviews with 12 students [31] showed that the use of libraries has diminished, and students tended to use the space individually. Researchers also emphasised the importance of space and furniture in relaxing students and keeping them in the library for longer after the pandemic. However, due to its qualitative and case study approach, this study was limited to the small sample size and generalizability. Similarly, how students choose library desks, which is influenced by their expectations of the indoor environment, may have also been affected. Investigation of changes in students’ seating behaviours and maintaining flexible environments to meet their new expectations may be crucial to keep them motivated and satisfied while avoiding unnecessary energy consumption. One of the significant factors affecting energy use in the built environment is occupancy behaviour, which may account for 64% of the difference between predicted and actual energy consumption in buildings [30].

## 3. Methods

### 3.1. Field Site

The study was carried out in the UCL Bartlett library, located on the ground floor of a six-story building. The library comprises three main study areas (Figure 1a). Room 1 has eight shared desks and four individual cubicles, Room 2 has twelve shared desks and eleven individual desks, and Room 3 has thirty-two shared desks. Regarding daylight, Room 1 has two north-facing side windows, and Room 2 has several side windows facing north and east orientations. Room 3 is an open-plan space with two skylights.

### 3.2. Occupancy Monitoring

The utilization of seats in the library has been monitored and recorded on a 10-min basis since 2017 [31]. The purpose of monitoring the occupancy in each UCL library (around 4000 seats) including the UCL Bartlett Library is to provide students with the real-time spatial distribution of available desks via an app called ‘UCL Go!’ (Figure 1b). The app provides real-time information on the availability of study spaces. The app enables students to find available spaces quickly and saves their time, especially during highly utilised periods, such as exam periods. It also allows students to choose an adequate study space according to their needs and expectations, which could considerably impact their academic performance [32]. Occupancy data is obtained from PIR sensors with infrared technology attached to each desk’s base (Figure 1c). Occupancy data is sent to OccupEye Cloud and is plotted using a range of red and green colours that indicate the percentage of time that desks have been occupied (Figure 1d). The collected data can be extracted daily, weekly, monthly and annually.

Since collecting data from individuals in the real world was not feasible, this study used motion sensors to track changes in students’ library usage over an extended period of time. The methodology is based on the assumption that occupancy monitoring data accurately represents the individuals’ seating preferences because an experiment in the Bartlett Library [33,34] and a subsequent occupancy monitoring study [11] both identified specific spaces as being the most occupied and most in-demand.

### 3.3. Occupancy Monitoring

This study used motion sensors to track changes in the use of the library over an extended period. The methodology is based on the assumption that occupancy monitoring data accurately represents the individuals’ seating preferences in the Bartlett Library [33,34] and a subsequent occupancy monitoring study [11] both identified specific spaces as being the most occupied and most in-demand. Therefore, the data obtained from the occupancy monitoring system was analysed to understand the role of COVID-19 in the seating behaviour of students. The analysis was carried out considering the utilisation of desks between 9:00 and 20:00 on weekdays and between 11:00 and 18:00 on Saturdays. The comparison of seating behaviour before and after the pandemic was based on data collected between October 2019 and March 2020, and between October 2021 and March 2022 as periods before and after the pandemic. The data was analysed in the following ways:**Desks and rooms with more and less demand:** The annual occupancy of each desk was analysed to investigate the desks and rooms with more and less demand before and after COVID-19.**Order of preference of desks**: The degree of freedom of choice could influence the seating decision because individuals can choose only available seats. For instance, students could have more chances to select desks early in the morning than students arriving in the afternoon. Thus, the selection of desks in the morning hours was analysed on weekdays from 9:00 to 12:00 at 30 min intervals for a month as presented in Table 1. The 30-min time interval was defined because students on average occupy the desks for at least 30 min in the morning. In order to investigate which desks were preferred earlier than others on a typical day, the percentage of time a desk was occupied between 9:00 and 12:00 was calculated. The analysis was limited to noon because the library reaches the first peak of occupation at midday on weekdays. As seen below, a desk’s occupancy rate on a specific date and time is calculated with the ratio of occupied cases to total cases throughout the month. If a desk was occupied at equal to or more than 90% of the month (acceptable confidence interval in studies with small sample sizes), then that desk was regarded as occupied. As an example, Table 1 illustrates how the utilization of Desk 1 was calculated for a month and this desk was typically utilised at 13.8% between 9:30 and 10:00 in October 2019. Therefore, this information could be used to compare the utilization of this desk in relation to others within a time frame.

The Bartlett library has various layouts which can be considered when assessing students’ seat preferences. Previous research has shown that the decision on seat selection arises from factors; such as daylight [27,28], ambient temperature, type of furniture, proximity to other occupants [25], quietness, outdoor view, privacy, social interactions such as close to friends, entrance or circulation [26], students’ degree of territoriality and seat arrangements [29]. Figure 2 and Figure 3 show and describe the zones defined and considered some of the factors mentioned above.

## 4. Results

### 4.1. Change in the Number of Students Visiting the UCL Bartlett Library

Table 2 illustrates the average utilisation of the library from pre-pandemic to post-pandemic. In the UK, a national lockdown was introduced on 23 March 2020 and lifted after September 2021. As a result, usage of the library was extremely low between April 2020 and October 2021. The number of students visiting the library decreased dramatically with the lockdown. The library was used at nearly two-thirds of the pre-pandemic rate, and students have not yet resumed their previous usage since the restrictions were lifted. It seems that students do not spend time indoors as they used to before the pandemic, or perhaps they avoid staying in the library when it becomes highly occupied. Interestingly, the library was used less than half as much during the summer closure period (May to July 2022) after the pandemic compared to 2019.

### 4.2. The Variation in the Utilisation of Desks before and after COVID-19

Figure 4 shows the average utilisation of each desk during pre- and past-pandemic periods. Approximately 81% of desk utilisation after the pandemic could be explained by pre-pandemic utilisation (*p* = 0.05). Although overall utilisation of the library decreased, the pattern in desk occupancy resembles that prior to the pandemic. However, while some desks are still preferred, others are not selected as previously. Thus, the variation in desk utilisation before and after the pandemic highlights the importance and necessity of this study.

### 4.3. The Variation in the Utilisation of Desks before and after COVID-19

Before the pandemic, Room 2 which has twelve shared and eleven individual desks was the most popular in the library (60.2%). The utilisation of Room 1 and Room 3 was almost the same at 37.5% and 38.0%, respectively. Although all rooms have been less utilised after COVID-19, Room 2 still has the highest occupancy at 47.5%. Previous research [11,33,34] revealed that the reason for the high selection of desks in this room is the accessibility to daylight and outdoor views.

Although the demand from students was almost the same for Room 1 and Room 3 before the pandemic, Room 1 now has a higher occupancy (28.4%) than Room 3 (23.1%). Room 1 is smaller in size, but it has fewer shared desks (8) with access to outdoor views and daylight, as well as four individual cubicles that provide students with privacy and some sort of separation from others. Room 3, on the other hand, has more shared desks (32) in an open-plan space with access to daylight provided by two skylights but no outdoor view.

### 4.4. The Effect of the Zone’s Characteristics on Students’ Seating Behaviours

The areas within rooms were classified based on the characteristics of desks and layouts, as shown in Figure 2. Table 3 shows the utilisation of the defined zones before and after COVID-19. Although overall library usage has decreased, the difference in utilization before and after the pandemic was regarded as an indicator of the popularity of those desks and zones. Hence, the lower difference indicates greater popularity. The four individual cubicles in Zone B located in Room 1 have the least difference in utilization. Following that, three hot desks in Zone E remain popular as before although they are located on a circulation route and do not have access to daylight or outdoor views. Other desks presenting a low difference are the eight shared desks in Zone A, located along two side windows. Although these are shared desks, they are separated by desk dividers that provide privacy between students. Interestingly, the popularity of the eleven individual desks in Zone D decreased considerably, even though they were the most popular desks prior to the pandemic with access to outdoor views and daylight.

### 4.5. The Effect of the Desk’s Characteristics on Students’ Seating Behaviours

In addition to the overall utilisation of the library, the investigation of what types of desks are more in demand in the library and whether there is any change in the demand for spaces after the pandemic needs consideration. Figure 5 shows the difference in students’ seat selections after COVID-19. The most interesting difference in desk utilisation is that, while the desks next to the windows were well utilised in Zone C, desks next to them on the circulation route were not chosen as before. Similarly, while one desk had a higher occupancy, the opposite desk had a lower occupancy in Room 3. However, there was no discernible pattern in shared desk occupancy in Zone A. Furthermore, desks getting a high level of daylight (Zone F) were not as popular as they did before the pandemic. The corner desk in Zone E, which is comparatively more private than other desks in the same zone, continued to be utilised between 40% and 60% after the pandemic, however, the utilization of the other two desks decreased. Desks in Zone B, on the other hand, maintained the same occupancy pattern after COVID-19.

The degree of freedom of choice could also influence the seating selection of students because students can choose only available desks. For instance, they could have more chances to select a desk in the early morning than those arriving in the afternoon. Figure 6 and Figure 7 show the utilisation of desks in the early morning hours as occupied and unoccupied, before and after COVID-19, respectively. 

As seen in Figure 6, first comers to the library mostly prefer the individual desks in Room 2. These desks have a good combination of daylight, outdoor view and privacy. Following, students seem to prefer the shared desks in Room 2 with an outdoor view and comparatively less daylight availability and less privacy. After the desks in Room 2 were fully occupied (between 10:00–10:30), students initially selected desks in other rooms, mostly those with a high level of daylight and far away from other students as much as possible. When the desks getting a high amount of daylight in Room 3 are fully occupied, students begin to select other desks in the same room with less daylight. 

Figure 7 shows the seating selection of students in the early hours after COVID-19. As seen, first comers to the library still mostly prefer the individual desks in Room 2. At the same time, some students prefer the shared desks at corners in Room 2 with an outdoor view and comparatively less daylight availability and less privacy. Differently from the seat preference before the pandemic, students select desks from other rooms before previously preferred ones in Room 2. The desks initially chosen from other rooms were mostly in the corners and those apparently isolated regardless of access to daylight or outdoor view. One of the most interesting findings from this analysis was that shared desks next to the circulation route in Zone C were not chosen as before whereas desks next to the window were fully occupied. A similar situation was seen with the shared desks in Room 1 and Room 3. Students seem to prefer shared desks when desks either next to or opposite to them are empty.

In addition to comparing the seating selection of students before and after the pandemic, the occupancy pattern from June 2022 to August 2022 was also analysed to investigate if students have completely resumed their use of libraries. However, the study results showed that the occupancy rate of each desk remains similar to occupancy patterns observed in the months just after the pandemic, October 2021 to March 2022 (*p* = 0.05). This finding demonstrates that change in students’ seating behaviour is not temporary and needs further attention. 

## 5. Discussion

### 5.1. Change in the Number of Students Visiting the UCL Bartlett Library

Findings showed that the number of students visiting the library decreased dramatically with the lockdown. Since the restrictions were lifted, students have not yet resumed their previous library usage, and the library has been used at nearly two-thirds of its pre-pandemic rate. A possible explanation might be that students are not willing to spend more time indoors as they used to before the pandemic, because they had already been restricted to staying at home for a long time. This is in line with a study where changes in students’ use of libraries following the pandemic were investigated [35]. Although no questions were asked in this study, nearly half of the participants independently mentioned the benefits of using outdoor seating, and they found those places convenient for studying and, relaxing, and allow for enjoying views and natural light. The reduction in the number of students using the library might also be explained by the students’ concerns about spending time in highly occupied indoor spaces due to the still-existing risk of COVID-19 transmission. This also accords with a previous study, which showed that restricting people’s mobility and social isolation to minimize the spread of the virus during the COVID-19 pandemic can put a significant strain on people’s mental health on a scale unprecedented in recent history [36]. Thus, people might still be afraid of contacting individuals possibly infected by COVID-19. Finally, the decrease in the number of students using the library might be related to students who have gotten used to working from home and might not be willing to come to the library as pre-pandemic because the idea of a workplace has changed completely [37]. Surprisingly, during the summer closure period following the pandemic, the library was used nearly half as much as it was in 2019. This result may be related to students avoiding staying indoors when there is no obligation for studying indoors during the closure period because they had already been forced to stay at home for a long time. 

Maintaining sufficient and well-utilised study spaces is crucial for avoiding high energy expenditure as well as maintaining students’ satisfaction with the library environment. Therefore, managers could set up new environmental control systems that match actual building occupancy more closely than current settings. Even some rooms could be closed during periods of low occupation to reduce unnecessary operational energy consumption. They can also redesign the layout of the desks based on students’ needs and expectations.

### 5.2. The Variation in the Utilisation of Desks before and after COVID-19

One interesting finding is that some desks are not preferred and selected as there were before the pandemic (Figure 4). This result may be explained by the fact that individuals’ perceptions and usage of spaces were changed with the pandemic. Many studies have reported that COVID-19 caused a change in individuals’ relationships with space. For example, before the pandemic, visiting urban green space was considered non-essential; however, during the pandemic, it became an essential activity [38]. Another study also highlighted the changes and differences in the frequency and use of public spaces before and after pandemic restrictions [39]. However, the change is not limited to isolation periods; it is also predicted to affect individuals’ future use of space [7]. The changes in the perception and use of spaces might also be related to changes in the feelings towards environmental factors in the buildings [40] because various studies demonstrated that the exposed environment during the isolation period affected individuals’ needs and expectations from the indoor environment. For instance, a study demonstrated the changes in human perception of environmental sounds in an urban neighbourhood during the COVID-19 lockdown, pointing out the effect of a reduction in traffic noise and an increase in birdsong. The perceptional change was linked to the changes in the soundscape environment to which individuals were exposed during the COVID-19 lockdown [41].

### 5.3. The Effect of the Room’s Characteristics on Students’ Seating Behaviours

One of the most interesting findings is that after the pandemic, Room 1, despite its smaller size, has a higher occupancy (28.4%) than Room 3 (23.1%). This result, however, conflicts with a study that found that open-plan spaces in an academic building were much more efficiently utilised than enclosed spaces [35]. Even after the pandemic, students appear to be concerned about keeping their distance from others. This behaviour could be explained by the different seating arrangements or the ventilation provision in these two rooms. Students might not choose to select a desk in Room 3 because it has an open layout forcing students to sit face-to-face, increasing the risk of COVID-19 transmission. The difference in window control systems in these two rooms may explain why students prefer desks in Room 1 despite the room’s smaller size because students can get fresh air from the two side windows in Room 1 when they need it, whereas UCL managers control the skylights in Room 3. It is now well known that opening windows is the simplest way to improve ventilation in a room and prevent the transmission of infectious diseases like COVID-19.

### 5.4. The Effect of the Zone’s Characteristics on Students’ Seating Behaviours

One unexpected finding was that the popularity of the eleven individual desks in Zone D decreased significantly, despite the fact that they were the most popular desks with access to outdoor views and daylight prior to the pandemic. Although these are individual desks, their low utilization after the pandemic may be explained by the fact that they have shorter dividers than those in Room 1. Shared desks in the same room (Zone C) were also used less (16.1%) than before COVID-19. This decrease in the demand for those desks could be their short desk dividers and the simple fact they have to be shared with other students.

Zones F and G located in an open plan space with 32 shared desks showed the greatest decrease in desk usage (17.3 and 15.6%) following the pandemic. A possible explanation for this result may be related to the position of desk dividers on the desks. In those zones, desk dividers are placed between desks, as opposed to other rooms where they are placed in front of the desks, preventing students from facing each other. The number of students sharing the same room may also be a reason to avoid selecting these desks; because crowded settings can raise the likelihood of being close to someone with COVID-19 where people tend to spend longer periods of time [42]. 

Surprisingly, the greatest difference in utilisation within zones was observed in Zone F, even more than in Zone G, despite the fact of excellent daylight levels. It demonstrates that COVID-19 significantly influenced student seating behaviour, as most stated reasons for seat selection prior to the pandemic (e.g., daylight and outdoor view) have become less important now.

### 5.5. The Effect of the Desk’s Characteristics on Students’ Seating Behaviours

As seen in Figure 5, desks next to windows in Zone C were well used, but those on the circulation route were not as frequently chosen as before. This difference could be related to the students’ desire to maintain distance and remain as isolated as possible, so when desks next to the windows are occupied, those next to them stay empty. In Room 3, a similar pattern to Zone C was observed; while one desk had a higher occupancy, the opposite one had lower. This is most likely due to students’ avoidance of sitting facing other students without the apparent protection a desk divider may provide.

After the pandemic, corner desks became preferable most probably they were comparatively more private than others (Figure 6 and Figure 7). Although the desks near the windows in Room 1 have access to daylight as much as some desks in Room 2 and have a similar outdoor view, those were not preferred by students firstly. It could be explained that Room 1 has a North orientation and is comparatively darker than Room 2, especially in the early morning hours. Another reason that could explain this situation might be the room size. Furthermore, the individual cubicles facing the wall in Room 1 were selected earlier than the shared desks by the window, probably due to privacy and/or being afraid of contacting individuals possibly infected by COVID-19. 

Together these findings proved that students’ seating selection noticeably changed following the pandemic. Students appear to prefer isolated desks or desks that provide a certain distance from other students. This situation might be problematic for students’ health and well-being and productivity in the future. When students choose desks solely to be isolated from others, without consideration of other factors mentioned as reasons for seat selection prior to the pandemic, such as daylight availability, outdoor view, quietness, and so on, their health and well-being might be affected adversely and they might not be as productive as they were. On the other hand, a reduction in the number of students visiting the library is critical not only for the effective use of libraries but also this reduction highlights the decrease in the number of students who come together in the library environment which has great importance, especially after a long period of isolation. 

### 5.6. An Architectural Element Influencing Students’ Seat Selection after COVID-19

Students’ seating preferences appear to be also influenced by the position of desk dividers which were already in the library before the pandemic. Table 4 shows the most likely seating patterns in different configurations of seating layouts and positions of desk dividers. In this study, three different patterns were noticed. Firstly, it was noticed that students avoid seating side-by-side when the dividers are placed in front as in Layouts 1, 3 and 4. In this case, students tend to sit face to face or staggered when dividers block their connection with the opposite person, and there is no barrier between the person next to each other. Secondly, students avoid seating face to face when the dividers are placed between students seated side-by-side, as in Layout 5. It was noticed that students tend to sit staggered, with double dividers in between, or side by side when there is no divider in front. Thirdly, students tend to sit anywhere where a divider entirely protects a desk as in Layout 2. The height of the dividers may also have a role in students’ seating selections because a noticeable difference in seating preference of students was observed in Zone A and Zone C, where desks have a similar layout and position of desk dividers. Students do not avoid sitting face-to-face in Zone A, where the dividers are higher than those in Zone C. On the other hand, students prefer more staggered seating in Zone C, where the dividers do not completely block eye contact. 

Students’ belief to be protected from COVID-19 with desk dividers might be related to exposure to physical barriers or partitions that were nearly ubiquitous in public indoor spaces such as markets, hospitals, etc. during the pandemic. However, it is interesting that their seating choices, whether intentional or unintentional, correspond to the findings of a study investigating the effectiveness of various types of desk dividers on COVID-19 virus transmission [43].

As an architectural element in the built environment, desk dividers seem to have a role in seat selection as a safety mechanism. The influence of desk dividers on students’ seat selection could be related to either privacy or a prevention method for COVID-19, however the question of why a similar pattern was not seen before the pandemic supports the idea of protection against infectious disease transmission. Future designers should consider not only the layout of the library desks but also the position, design and characteristics of the desk dividers.

### 5.7. Limitations, Future Research and Implications

This study’s main limitation was that occupancy is not always based on human presence. Students occasionally leave their laptops, water bottles, and backpacks to claim a seat while they go outside. PIR sensors can not detect a claimed seat due to no large heat signatures or movement. This situation could affect students’ freedom of choice and, ultimately, the interpretation of seat selection of students. This is because motion sensors assume that the desk is available in the case of no movement, but in reality, the desks could have been occupied with students’ belongings, not allowing others to select them. In order to avoid this limitation, the researcher should take a snapshot of spaces at regular time intervals with either photo or video recording.

The main contribution of this study is to increase our understanding of how COVID-19 has affected students’ seating behaviours in the library environment, as a result of changes in their needs and preferences from the indoor environment. Strategically investigating the students’ needs for spaces in the library environment is required to provide appropriate spaces through furniture reconfiguration, space assignment, and re-zoning the spaces such as collaborative, quiet zones, and so on. Therefore, the findings of this study could be used to understand how the layout of the spaces in library buildings could be improved to create a more pleasant workspace and applied to develop strategies by managers and daily operators of university buildings to improve the quality of libraries with the changing use of libraries. The re-arrangement of libraries will not only improve the mental health and productivity of students but setting up a new layout design that matches actual building occupancy more closely than current settings could also help to ensure the libraries are used more efficiently and ultimately, reduce unnecessary energy consumption in libraries. Future designers should also consider not only the layout of the library desks but also the position, design and characteristics of the desk dividers.

## 6. Conclusions

In the built environment, COVID-19 and the required physical distancing caused uncertainty about how this disease will affect occupants’ perception and use of space and how the design of public spaces will need to be adapted. Our understanding remains also limited in how students using libraries would react to the restrictions to limit transmission and how they would resume their activities when those restrictions are lifted. 

The main purpose of this research paper was to investigate the changes in how study spaces were used and what type of spaces were most in-demand before and after the pandemic. This study has three important findings: Firstly, it was found that COVID-19 impacted the utilization of the UCL Bartlett Library and the number of students visiting the library. Secondly, the findings revealed that COVID-19 seems to have affected student behaviour and the selection of desks. Students appear to prefer isolated desks or desks that provide a certain distance from other students, hence, ignoring parameters such as daylight and outdoor views, which were found highly important before the pandemic. The final finding relates to the existing desk dividers and their role in reducing COVID-19 transmission. A detailed study on students’ seat selections in other library configurations is required to understand the role of COVID-19 in their seating selection and to determine whether students are likely to use libraries as they used before the pandemic. It is also recommended that when designing a library, designers consider not only the layout of the desks but also the position, design and characteristics of the desk dividers to ensure the privacy of students and minimise infectious disease transmission.

## Figures and Tables

**Figure 1 ijerph-19-13255-f001:**
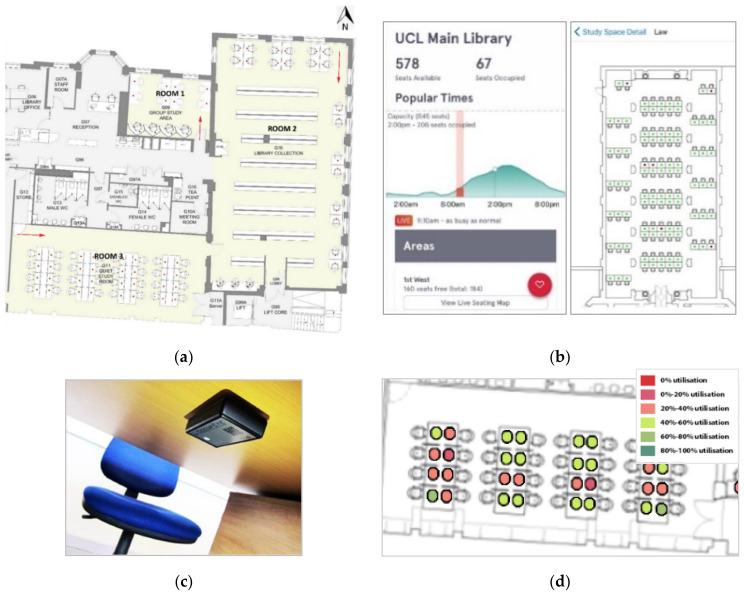
(**a**) Plan of the UCL Bartlett Library (**b**) Space availability information obtained from the UCL Go! Application (**c**) PIR sensors and (**d**) representation of occupancy at each seat in Occupeye Cloud.

**Figure 2 ijerph-19-13255-f002:**
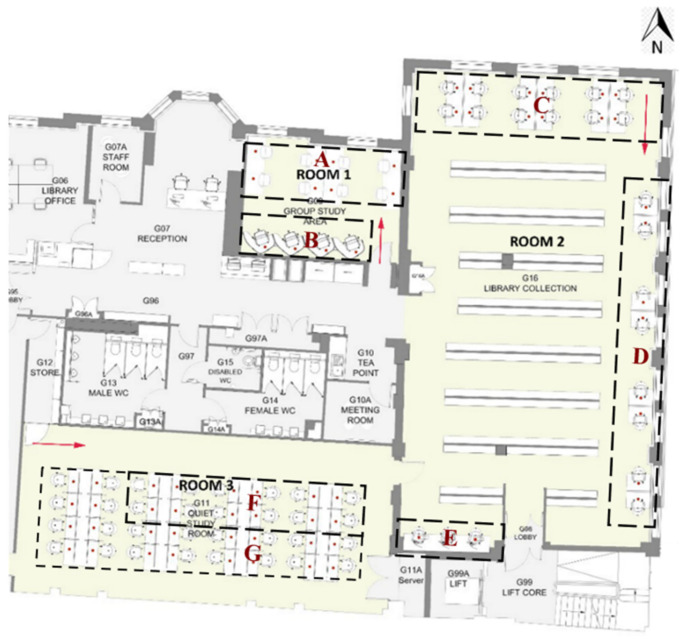
The zones with the common features in the library.

**Figure 3 ijerph-19-13255-f003:**
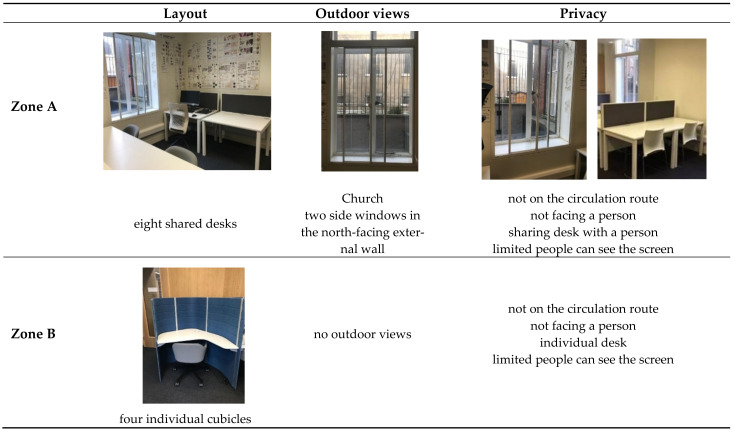
Defined zones in the Bartlett library.

**Figure 4 ijerph-19-13255-f004:**
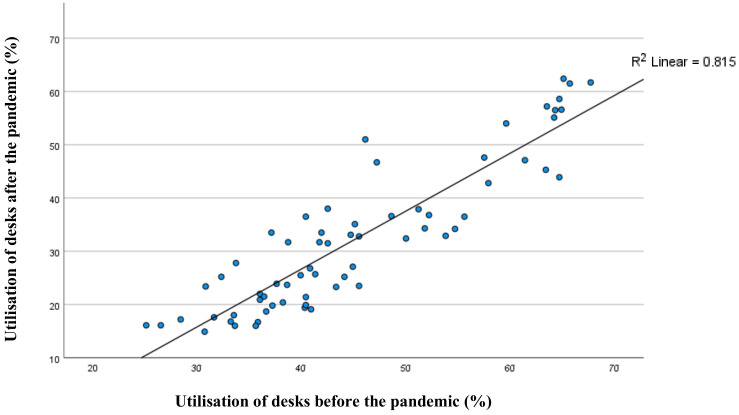
Comparison of desks’ utilisation between pre and post COVID-19.

**Figure 5 ijerph-19-13255-f005:**
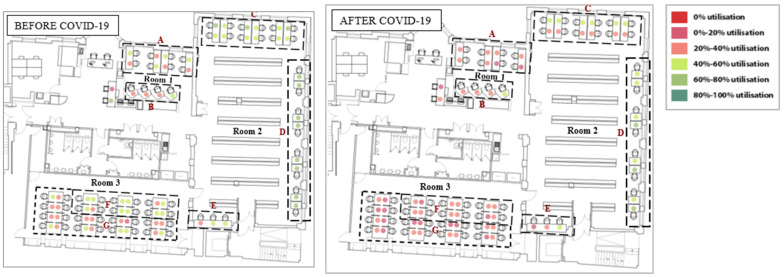
Change in the utilisation of the desks at the UCL Bartlett Library before and after the COVID-19 pandemic.

**Figure 6 ijerph-19-13255-f006:**
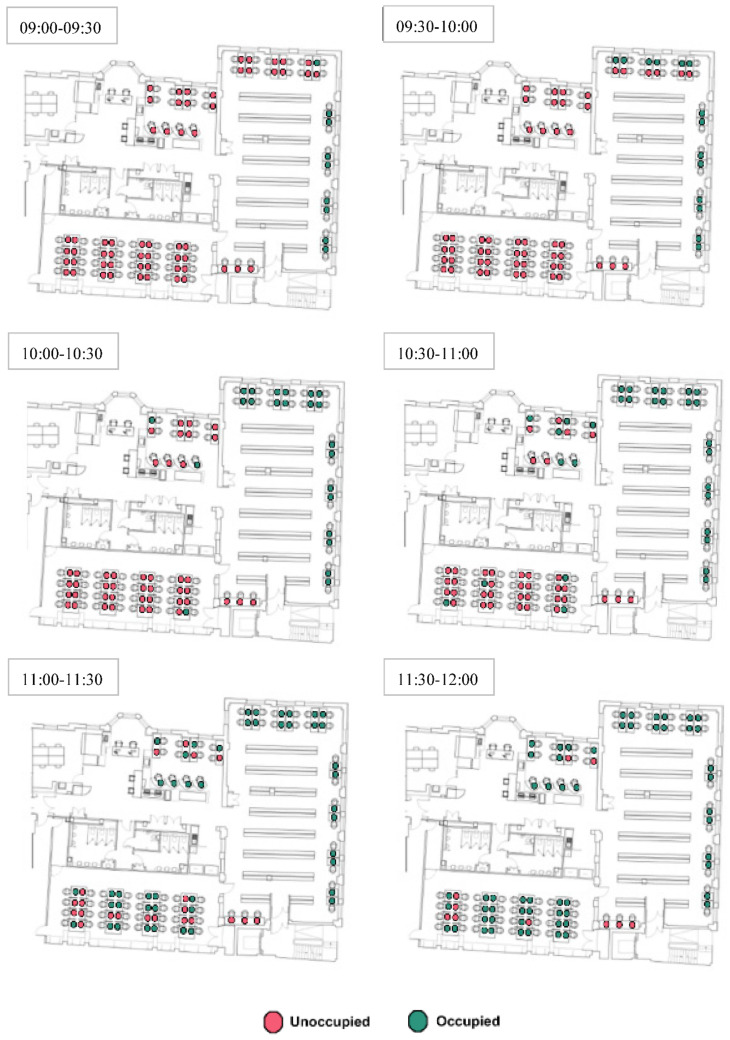
Seating preferences of the students in the early hours before COVID-19.

**Figure 7 ijerph-19-13255-f007:**
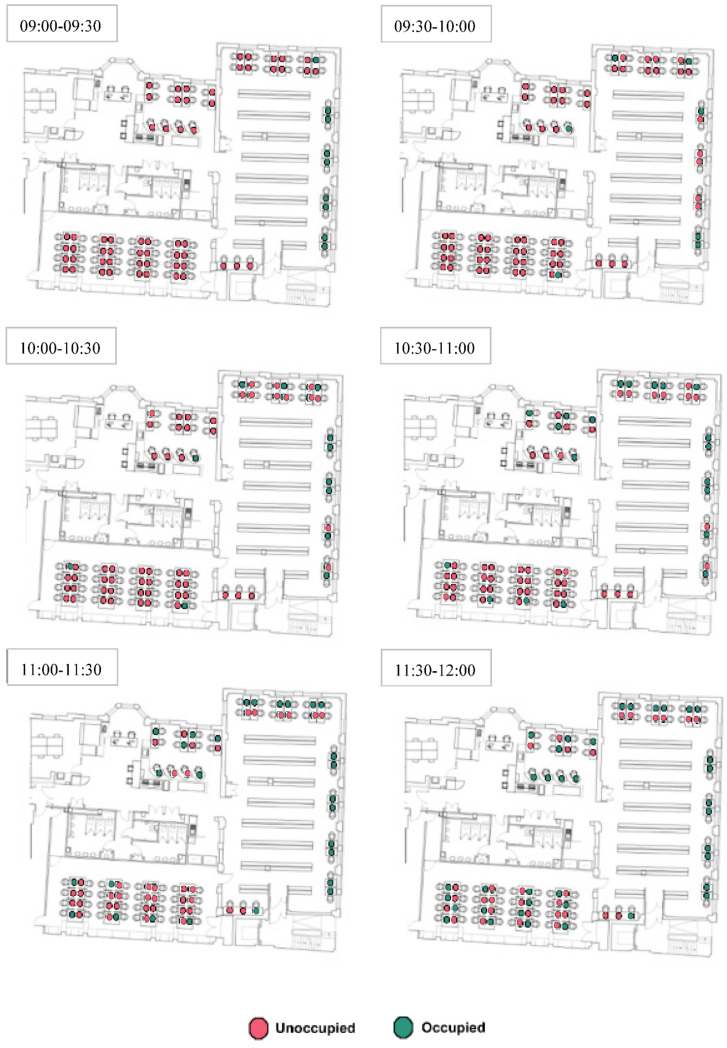
Seating preferences of the students in the early hours after COVID-19.

**Table 1 ijerph-19-13255-t001:** Method of analysis for monthly occupancy averages for specific time intervals (1: occupied, 0: unoccupied).

	Time of the Day					
Date	9:00–9:30	9:30–10:00	10:00–10:30	10:30–11:00	11:00–11:30	11:30–12:00
1 November 2019	1	1	1	1	0	1
2 November 2019	1	0	1	0	1	0
3 November 2019	0	0	0	1	0	1
⋮ ⋮	⋮ ⋮	⋮ ⋮	⋮ ⋮	⋮ ⋮	⋮ ⋮	⋮ ⋮
30 November 2019	1	0	1	1	0	0
Occupancy	10%	13.8%	35%	48%	57%	82%

**Table 2 ijerph-19-13255-t002:** Average utilisation of the library since 2019 (%) before, during and after COVID-19.

	Jan	Feb	Mar	Apr	May	June	July	Aug	Sep	Oct	Nov	Dec
2019	58.1	58.4	65.4	54.6	50.2	42.5	42.8	46.0	21.5	52.6	54.8	33.7
2020	48.5	54.1	21.9	0.1	0.0	0.1	0.0	0.3	1.3	7.0	4.9	2.6
2021	0.1	0.5	0.6	3.6	8.1	9.2	9.6	9.3	5.7	27.7	36.8	20.5
2022	27.8	38.0	41.3	34.3	30.9	20.1	17.2	18.1				

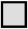
 Before COVID-19, 
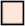
 During COVID-19, 
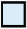
 After COVID-19.

**Table 3 ijerph-19-13255-t003:** Average utilisation of desks in the defined zones (%) before and after COVID-19.

	Utilisation before COVID-19 (%)	Utilisation after COVID-19 (%)	Differences in the Utilisation between before and after COVID-19 (%)
Zone A	37.1	27.5	9.6
Zone B	38.9	36.0	2.9
Zone C	57.0	41.9	15.1
Zone D	64.8	54.2	10.6
Zone E	38.7	33.5	5.2
Zone F	41.2	23.9	17.3
Zone G	36.0	20.4	15.6

**Table 4 ijerph-19-13255-t004:** Type of desk dividers in the library and the most likely seating patterns.

Desk Dividers		The Layout and Seating Pattern
Room 1—Zone A (h = 50 cm)	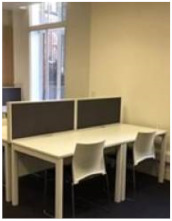	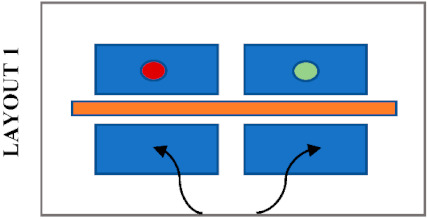
Room 1—Zone B (h = 80 cm)	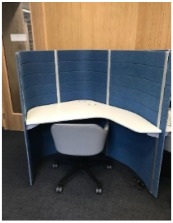	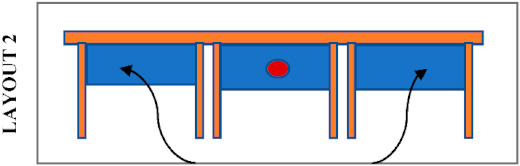
Room 2—Zone C (h = 40 cm)	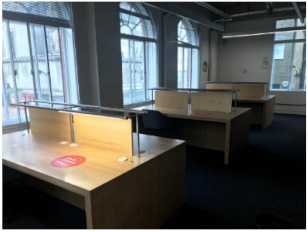	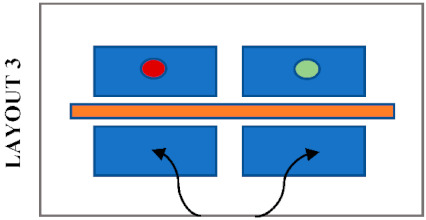
Room 2—Zone D (h = 40 cm)	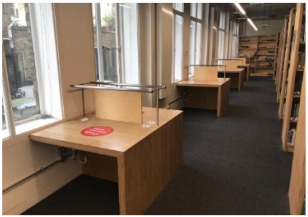	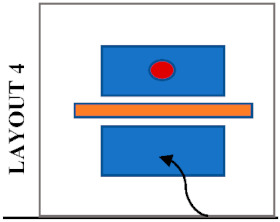
Room 3—Zone F and G (h = 40 cm)	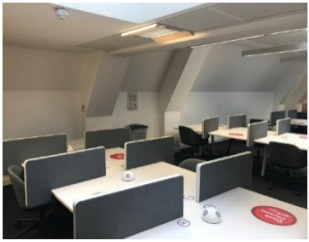	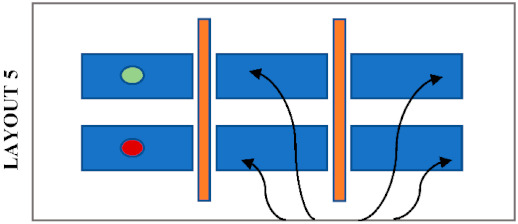
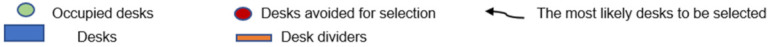

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
