# Peer review of "Seating Behaviour of Students before and after the COVID-19 Pandemic: Findings from Occupancy Monitoring with PIR Sensors at the UCL Bartlett Library"

_ijerph, 2022, doi:10.3390/ijerph192013255_

Round 1

Reviewer 1 Report

Dear Authors,

In my opinion, the article after correction could be sent to a journal with a different profile. Some of the doubts that appeared after reading the work can be found below.

The first paragraph of the introduction is too vague and does not provide any scientific basis for further discussion.

Please change the first sentence - either in the introduction or in the summary - to avoid repetition

Verse 32 - What does it mean that "the cities have been closed?"

Verse 34 - I will not agree that "everyone is limited to wearing masks inside and out" - all countries in the world have introduced such restrictions? In addition, in some countries there were groups of people who were exempt from this obligation for health reasons.

Please provide a more precise reference to the literature; the WHO website is too general;

There is also no indication from where the recommendation of social distance at a minimum distance of 1m came from ?? 1 meter is far too short a distance to prevent the virus from spreading via droplets; all the recommendations I knew said a distance of at least 2.5 meters.

Lines 36-41 - what exactly has changed? On what plane? How has "the lives of individuals" changed? How have habits changed?

Purpose of the work - is unclear, i.e. what does the work bring new to learning? And how does it affect public or individual health?

 References need to be supplemented: items 1, 15 and 16 are incomplete.

Conclusion:

The work requires many corrections, and considering the subject and purpose of the work (without justifying the impact on health), it should be submitted to a journal with a different profile (e.g. I suggest choosing a journal related to architecture or, for example, social behavior) - similar to the quoted literature.

There are far too few references (supported by literature) on health or environmental effects.

Author Response

Dear reviewer,

We sincerely appreciate your valuable time and insight to strenght to our paper. We made all possible changes you suggested and you can find the revised manuscript with track changes in Word format and Pdf format attached. We hope that you find it satisfactory and good enough to get published. 

Reviewer 2 Report

The authors have really done well in this research study. I have just observed some minor issues, otherwise ,the document is quite well written and has the potential for publication. Here are my suggestions to improve the overall quality of this document prior to publishing

Best Wishes

Title: Seating behaviour of students in the UCL Bartlett Library 2 before and after COVID-19

Abstract: well written

Introduction: it is quite vague in the current version of this manuscript why this study was conducted and what are its contributions to advance the existing body of knowledge ,and what kinds of benefit it has for the field.

Similarly, the knowledge should be explicitly stated in the introduction part.

I think the introduction should also suggest the structure of this document.

Literature: It is suggested to add more literature in this part highlighting the importance of this survey and comparing it by previous researchers. Currently, though the authors have done a good job, however, they have just presented a general overview of the current literature. Please include some more and relevant literature in this part.

Method and results : No major objection was observed.

Discussion:

Please discuss your results by comparing it with previous studies.

Similarly, social and theoretical implications of you study should be there before conclusion section

Moreover, please add limitation and future research direction.

Author Response

(The authors gave the same response as above.)

Reviewer 3 Report

Dear Authors, 

Despite being one case-specific study, the manuscript is very interesting and is going toward being accepted. I present a simple consideration. I recommend that the results section be separated from the discussion. Much information is mixed, which is undesirable for good reading and discussion quality. I also suggest a better organization of the figures, which in addition to having unused figures, appear to be outside of the recommended formats.

Nevertheless, this work is almost good. Try to create a better organization. After that, I think the work may be approved.

Author Response

(The authors gave the same response as above.)
